# Insights into Iron Metabolism Parameters in Ischemic Stroke: A Single-Center Prospective Cohort Study

**DOI:** 10.3390/ijms25179352

**Published:** 2024-08-29

**Authors:** Joanna Boinska, Artur Słomka, Magdalena Sury, Małgorzata Wiszniewska, Ewa Pisarek, Ewa Żekanowska

**Affiliations:** 1Department of Pathophysiology, Faculty of Pharmacy, Collegium Medicum in Bydgoszcz, Nicolaus Copernicus University in Toruń, 9 Marii Curie-Skłodowskiej Street, 85-094 Bydgoszcz, Poland; artur.slomka@cm.umk.pl (A.S.); zorba@cm.umk.pl (E.Ż.); 2Neurological Department with Stroke Unit, Specialist Hospital, 64-920 Piła, Poland; magda782@op.pl (M.S.); mwiszniewska@ans.pila.pl (M.W.); 3Faculty of Nursing, Stanisław Staszic State University of Applied Sciences, 64-920 Piła, Poland; episarek@ans.pila.pl

**Keywords:** ischemic stroke, thrombolysis, hepcidin, hemojuvelin, iron metabolism

## Abstract

The hemojuvelin–hepcidin regulatory axis may play a key role in the iron metabolism both systemically and locally. There is a pressing need to evaluate this tightly regulated network of iron parameters and their potential impact on the development of ischemic stroke (IS). We aimed to assess iron metabolism biomarkers in patients after IS, evaluating changes over time and considering their clinical features. We studied 45 patients diagnosed with IS. We assessed major iron metabolism parameters, such as hepcidin, soluble hemojuvelin (sHJV), soluble transferrin receptor (sTfR), and ferritin, using immunoenzymathic methods at two time points: on admission and on the 7th day post IS. We found increased ferritin levels on the 7th day post IS compared to admission, and this was observed in the entire study group (*p* = 0.03) and in the subgroup treated with thrombolysis (*p* = 0.02). The hepcidin levels, on the other hand, showed a significant decrease on the 7th day, though this difference was only evident in the entire study group (*p* = 0.04). We also discovered significantly elevated sHJV levels in patients with PACI stroke compared to other stroke locations, both on admission and on the 7th day post IS (*p* < 0.05). Significantly higher sHJV levels were observed in patients treated with thrombolysis compared to those receiving conventional treatment, regardless of the time point (*p* < 0.0001 and *p* = 0.0002, respectively). Our study revealed changes in the iron metabolism parameters during stroke. The patients with anterior cerebral infarction and those treated with thrombolysis presented significantly elevated sHJV levels.

## 1. Introduction

Stroke is a heterogeneous disease, ranking among the three major causes of death worldwide and frequently resulting in long-term disability. According to global epidemiological statistics, ischemic stroke (IS) accounts for 75% to 85% of all strokes [1,2]. The pathomechanism of IS is complex and influenced by both non-modifiable factors, such as age, sex, and genetics, and modifiable factors, including hypertension, hypercholesterolemia, atherosclerosis, and atrial fibrillation. However, regardless of the cause, thrombus formation disrupts blood flow, and the affected brain area becomes ischemic. For this reason, thrombolysis and thrombectomy are thought to be the best proven and effective therapeutic methods to restore cerebral blood flow and facilitate nerve function recovery [3,4].

In recent years, experimental and clinical studies have revealed a connection between IS and the iron metabolism. Iron overload can induce oxidative stress, leading to damage to proteins and the DNA. The production of reactive oxygen species (ROS) and lipid peroxidation associated with iron overload can adversely affect microglial cells and contribute to neuronal cell death through the process of ferroptosis [5,6,7,8]. When the blood–brain barrier is disrupted, excess iron can exacerbate neurological damage [9,10]. IS induces a cell–molecular response in many systems, aimed at developing the inflammatory response. Activated brain cells and vascular cells release proinflammatory cytokines that strongly affect iron homeostasis [11,12,13,14].

The tightly regulated network of proteins that regulate the iron metabolism in the brain is not yet completely understood. Hepcidin is believed to be a key regulator of systemic iron homeostasis. It is synthesized predominantly in the liver and regulates both iron absorption in the small intestine and the iron efflux from hepatocytes to the bloodstream by binding to ferroportin 1 (FPN1), a major iron exporter. Macrophages also play a critical role in this process, as they contain ferroportin and are involved in recycling iron by phagocytizing and degrading senescent and damaged erythrocytes. This recycling provides a continuous supply of iron for the production of hemoglobin in new erythrocytes and helps maintain systemic iron balance [15]. However, recent studies indicate that hepcidin mRNA is also present in various brain regions, including neurons and glial cells [16,17]. The regulation of hepcidin is influenced by several pathways, including erythropoiesis, inflammation, and iron signals. Among these, the iron signaling pathway is the most intricate and involves the careful collaboration of bone morphogenetic proteins (BMPs), transferrin receptor 2 (Tfr2), the hemochromatosis gene (HFE) protein, neogenin, and hemojuvelin (HJV) [18,19]. The HJV protein acts as a membrane receptor that facilitates the activation of the BMP-SMAD signaling pathway, ultimately leading to an increase in the hepcidin levels. However, soluble HJV (sHJV), which can be found in the bloodstream, functions as a negative regulator of the hepcidin gene by inhibiting BMPs [20].

Currently, there is limited information available regarding the iron metabolism parameters in patients with IS. It remains uncertain whether there is any correlation between the iron parameters and the etiology, location, severity, or type of treatment of the stroke. Furthermore, iron-related proteins may play a role in the processes of neuronal damage following a stroke and could have a significant predictive value for IS [21].

Thus, the aim of this study was to assess iron metabolism biomarkers in patients after IS at two time points, taking their clinical features into consideration.

## 2. Results

### 2.1. Patient-Specific Data

We conducted a prospective, single-center cohort study. A total of 45 patients (24 males) with a median age of 69 years, diagnosed with IS, were enrolled. Intravenous thrombolysis was administered to 25 patients (55.55%) according to national and international guidelines [22]. The remaining 20 patients (44.44%) received conventional treatment. Initially, we examined the clinical parameters and analyzed the differences based on the treatment method. Detailed characteristics of the study group are presented in Table 1. We found no significant differences between the two subgroups.

### 2.2. Iron Metabolism Parameters in Relation to Time Course and Treatment

Our main objective was to analyze the dynamics of changes in the iron metabolism parameters and compare the differences between subgroups based on the type of treatment they received. The ferritin levels exhibited an increase on the 7th day compared to admission, and this increase was observed in the entire study group (*p* = 0.03) and in the subgroup treated with thrombolysis (*p* = 0.02). The hepcidin levels, on the other hand, showed a significant decrease on the 7th day, though this difference was only evident in the entire study group (*p* = 0.04). Regardless of the treatment type, there were no significant variations in the levels of sTFR and sHJV between admission and the 7th day following IS (Table 2).

Notably, in terms of the iron metabolism, only the sHJV levels exhibited significant differences between the thrombolysis and conventional treatment subgroups, both upon admission and on the 7th day. Furthermore, the median values of sHJV in the thrombolysis subgroup were found to be twice as high as those in conventionally treated patients with IS (Table 2).

### 2.3. Analysis of Iron Metabolism Parameters Depending on Selected Clinimetric Scales

In the next step, the parameters of the iron metabolism were assessed on admission and on the 7th day post stroke, depending on clinical scales, to evaluate whether there is a correlation between these parameters and the severity, etiology, and location of the stroke.

#### 2.3.1. Stroke Severity (NIHSS)

The patients were categorized into three subgroups of IS severity based on the NIHSS: mild (<8 points), moderate (8–15 points), and severe neurological deficit (≥16 points). However, no significant differences were found in the concentration of the analyzed iron metabolism parameters among post-stroke patients, regardless of NIHSS severity (Table 3). Furthermore, a correlation analysis revealed no association between the concentration of the determined parameters and the NIHSS scores.

#### 2.3.2. Stroke Etiology (TOAST)

There was no significant impact of stroke etiology according to the TOAST classification on the examined iron parameters both on admission and on the 7th day of hospitalization. However, patients with IS of an unknown etiology had a significantly higher sTfR concentration on admission compared to those with a stroke caused by the atherosclerosis of large vessels (Table 4).

#### 2.3.3. Stroke Location (OCSP)

Considering that the pattern of the clinical picture and the severity of symptoms depend on the location of IS, we next evaluated the iron metabolism parameters according to the OCSP classification. A specific group of patients with anterior circulation infarct had considerably higher sHJV concentrations (Figure 1). Notably, significant differences in the sHJV levels were observed between partial anterior circulation infarct (PACI) and posterior circulation infarct (POCI) (both on admission and on the 7th day), as well as between PACI and lacunar infarct (LACI) (on admission). Table 5 reveals that there were no substantial variations in the ferritin, hepcidin, and sTfR levels based on stroke location.

## 3. Discussion

The discovery of hepcidin in 2001 and the HJV gene in 2004 significantly reshaped our understanding of iron homeostasis and its role in brain physiology [23]. Despite these advancements, much remains unknown about how iron dysregulation affects brain cells and contributes to neurological disorders. Our study addresses these gaps by evaluating various iron metabolism parameters, including hepcidin, sHJV, ferritin, and sTfR, in 45 patients with IS at two time points: upon admission and on the 7th day post stroke.

In the first step, we assessed changes in the iron metabolism parameters with time. Interestingly, the ferritin levels were significantly increased on the 7th day compared to admission in all the patients with IS and, specifically, in the thrombolysis subgroup. We observed a decrease in the hepcidin concentration on the 7th day compared to admission, although a significant difference was evident only in the whole study group. Nonetheless, the sTfR and sHJV concentrations remained unchanged across the entire group, regardless of the treatment type.

As part of this study’s research task, we looked into how different treatments (thrombolysis vs. conventional) affected the iron parameters after IS. The results showed that the sHJV concentration was more than double in patients who received thrombolysis on both admission and the 7th day of hospitalization. We found no differences in the hepcidin, sTfR, and ferritin concentrations with respect to the type of treatment implemented.

After conducting a thorough investigation, we discovered that the patients diagnosed with anterior circulation infarct (PACI and TACI) had the highest levels of sHJV. However, there were no significant differences in the hepcidin, ferritin and sTfR levels between the various groups classified by the OCSP. Moreover, we found higher sTFR values in the patients with cardiogenic or undetermined stroke. However, there was no correlation between the examined parameters and the intensity of neurological deficits according to the NIHSS.

According to the study results, the patients with strokes in the anterior vascular system and those who received thrombolysis had more than twice as high levels of sHJV compared to the rest, both before (on admission) and after alteplase administration (on day 7th of hospitalization). These results strongly support the findings in the work of Young et al. (2020), who conducted an extensive analysis of the role of HJV in ischemic stroke using a wide range of methods [21]. In their study, histological studies of brain tissue from patients with ischemic stroke showed significantly higher HJV expression compared to the brain tissue from patients without stroke. Similarly, the analysis of sHJV in the systemic circulation showed much higher concentrations in patients with ischemic stroke compared to healthy individuals [21]. This increase in the plasma concentration of hemojuvelin in patients during the acute phase of ischemic stroke was interpreted as damage to the blood–brain barrier and the systemic response to stress, resulting in the release of HJV into the systemic circulation from the liver and skeletal muscles.

We cannot answer the question of what stands behind the significant increase in sHJV in patients with anterior cerebral infarction. It is known that anterior circulation infarct stroke is usually associated with a poor outcome [24,25]. According to the mechanism of action of sHJV, elevated levels of sHJV are known to reduce hepcidin expression, which, in turn, increases the iron levels in the blood. The clinical usefulness of our findings remains an open question that requires further analysis in the context of prospective studies related to the functional assessment and determination of the cut-off point for the HJV concentration associated with a potential poor prognosis.

We can speculate that elevated sHJV levels may result from at least two factors. The first is the time from the onset of symptoms to treatment, which is shorter in the case of thrombolysis. The second is the treatment itself, which may partially contribute to maintaining elevated sHJV levels. The work of Fredriksson et al. suggests that tPA may have pleiotropic properties and influence the integrity of the blood–brain barrier, which could also explain the frequency of hemorrhagic complications associated with thrombolysis [26]. Increased sHJV levels, by inhibiting hepcidin and releasing iron, could contribute to this effect.

Our research observed that hepcidin concentration significantly decreases on the seventh day of hospitalization, regardless of the treatment administered during the acute phase of stroke. While our study did not directly measure inflammation-related indicators such as C-reactive protein, a decrease in hepcidin could be consistent with findings from the related literature suggesting a link between inflammatory processes and hepcidin regulation. In contrast, the ferritin levels increased on the 7th day, which may have reflected changes in the iron metabolism. The differences in the behavior of these acute-phase proteins over the 7-day post-stroke period suggest distinct rates of normalization and potentially different regulatory mechanisms, particularly with regard to hepcidin. It is also important to recognize that these two proteins have distinct roles in the iron metabolism: ferritin functions as an iron storage protein, while hepcidin regulates the availability of iron in the circulating pool.

Experimental research has revealed that there are two sources of hepcidin in the cerebral circulation: local synthesis by microglial cells and a pool from the systemic circulation. In an animal model study, mice with inactivated genes encoding hepcidin showed no iron overload in neurons under normal conditions. However, when inflammation stimulators were present, increased hepcidin expression was observed, which led to the excessive accumulation of iron in cells through the blocking of ferroportin. This accumulation could be crucial in initiating neuronal degradation by ferroptosis. Experimental studies suggest that reducing brain damage can be achieved by blocking excessive hepcidin synthesis in response to inflammation, hypoxia, or iron overload [27,28,29].

According to research conducted by Słomka et al., patients in the acute phase of ischemic stroke may experience an increase in the concentration of hepcidin in their bloodstream due to the inflammatory reaction, particularly in those who undergo thrombolysis. However, the study found that patients who undergo standard treatment and are administered low-molecular-weight heparin (LMWH) have significantly lower levels of hepcidin. This is likely due to the ability of heparin to prevent the synthesis of hepcidin in the liver [30]. The inhibitory effect of heparin on the expression of *hepcidin* has been demonstrated in studies by Poli et al., both in vitro and in animal models [31,32].

As previously noted in our research, patients with cardiogenic stroke or stroke of an undetermined cause had significantly higher sTFR values when considering the etiological factors of stroke. These findings are in line with the studies conducted by other authors [30,33]. Tang et al. also confirmed that an increase in hypoxia-induced factor 1, an early ischemia-activated factor, promotes the expression of sTfR, resulting in an increase in free iron [33]. However, there are also reports indicating no significant differences in the sTfR levels in the systemic circulation of patients with IS [34].

The research presented in this study primarily contributes to our understanding of the iron metabolism in stroke. Clearly, in the examined population of patients, there is a group of patients whose concentration of the soluble form of HJV is statistically significantly higher. The analysis of the clinical data indicates that these are patients with a stroke located in the anterior part of the vascular circle, qualified for thrombolytic therapy.

This work has many limitations, which certainly include a relatively small group of examined patients, the fact that the tests were performed using peripheral blood, which does not fully reflect the mechanisms taking place locally, and that the results obtained refer only to a 7-day observation period. Continuation of this study is required to determine whether changes in the regulators of the iron metabolism, such as hepcidin or HJV, could be applied in the future for potential clinical use.

## 4. Material and Methods

### 4.1. Study Population

This study was conducted among 45 patients, diagnosed with IS, qualified for treatment in the Neurological Department with Stroke Unit, Specialist Hospital in Piła (Poland) (Figure 2). The stroke was diagnosed based on WHO criteria and confirmed by a computed tomography (CT) scan performed immediately after the patient’s admission.

The exclusion criteria were as follows: hemorrhagic stroke, transient ischemic attack (TIA), kidney disease, liver failure, cancer history, acute infection, steroid therapy, or pregnancy. Patients underwent physical examinations upon admission and again on the 7th day of hospitalization. Stroke etiology was assessed using the Trial of Org 10,172 in Acute Stroke Treatment (TOAST) classification. The symptoms and location of the stroke were determined based on the Oxfordshire Community Stroke Project (OCSP) classification system. Upon admission and on the 7th day, the extent of neurological deficits was evaluated using the National Institute of Health Stroke Scale (NIHSS), while global disability was assessed through the modified Rankin Scale (mRS). The Barthel Index (BI) was used to determine patient independence and the need for nursing care.

The study group comprised 45 patients with IS, of whom 25 met the eligibility criteria for thrombolytic treatment with recombinant tissue plasminogen activator (rt-PA, alteplase) administered within 4.5 h of stroke onset. The remaining 20 patients received conventional treatment. The inclusion criteria for thrombolytic therapy were based on the guidelines from the American Heart Association and American Stroke Association [22]. These criteria included the following: age ≥ 18 years, clinical diagnosis of IS with significant neurological deficit, time from stroke symptom onset to treatment within 4.5 h; duration of symptoms ≥ 30 min without significant improvement prior to treatment, and exclusion of intracranial bleeding on CT or magnetic resonance imaging (MRI).

This study was performed based on the guidelines of the Declaration of Helsinki and approved by the Bioethics Committee of Nicolaus Copernicus University in Toruń, Collegium Medicum in Bydgoszcz, Poland (KB 694/2016).

### 4.2. Blood Samples and Laboratory Tests

Blood samples were collected twice: once upon admission (before treatment) and again on the 7th day after IS. Venous blood was drawn into tubes (Vacuette, Greiner Bio-One, Kremsmünster, Austria) containing a clot activator. The samples were mixed, centrifuged at 2000× *g* + 4 °C for 20 min, aliquoted into Eppendorf tubes, and stored at −80 °C until analysis.

### 4.3. Iron Metabolism Parameters

The serum hepcidin levels were measured by DRG Hepcidin 25 (bioactive) HS ELISA (DRG Instruments GmbH, Marburg, Germany). The detection limit of this assay was 0.153 ng/mL. The intra-assay coefficient of variation (CV) was between 0.8 and 4.1%, and the inter-assay CV was between 9.5 and 14.4%. The sHJV levels were assessed using the ELISA kit by Cloud-Clone Corp. (SEB995Hu, Katy, TX, USA). The lower detection limit for HJV was 12.7 pg/mL. The intra- and inter-assay CVs were <10% and <12%, respectively. The soluble transferrin receptor (sTfR) was measured using ELISA kits purchased from BioVendor-Laboratorni medicina a.s. (RD194011100, Brno, Czech Republic). The analytical sensitivity of the sTfR assay was found to be 2 ng/mL. According to the manufacturer’s instructions, the intra- and inter-assay CVs for sTfR were 3.4–4.3% and 5.5–7.0%, respectively. The ferritin levels were measured by Ferritin ELISA (DiaMetra Srl Unipersonale, Perugia, Italy). The detection limit of this assay was 0.04 ng/mL. The intra-assay CV was ≤7.5% and the inter-assay CV ≤ 6.1%.

### 4.4. Statistical Analysis

All statistical analyses were performed using Statistica 13.3 for Windows (Tibco Software Inc., Palo Alto, CA, USA). The normality of distributions was tested using the Shapiro–Wilk test. The Chi-square test was used to compare the distribution of frequencies. The Mann–Whitney U test, Kruskal–Wallis ANOVA, and Wilcoxon signed-rank test were employed for the comparison of non-normally distributed variables. The correlations were assessed using the Spearman rank correlation test. A *p* value of less than 0.05 was considered statistically significant.

## 5. Conclusions

Our study revealed changes in the iron metabolism parameters during stroke. Patients with anterior cerebral infarction and those treated with thrombolysis present significantly elevated sHJV levels.

## Figures and Tables

**Figure 1 ijms-25-09352-f001:**
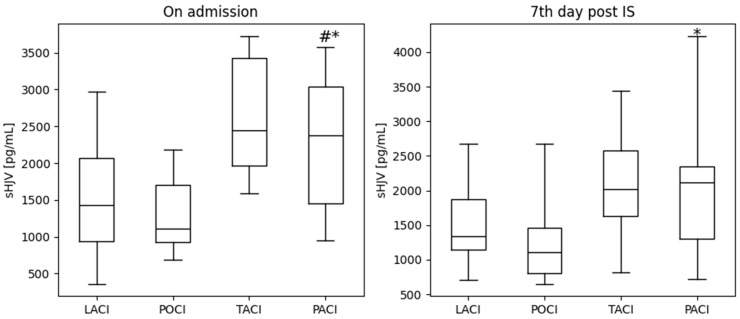
sHJV concentrations in relation to stroke location on admission and on the 7th day post AIS. * *p* < 0.05 compared to POCI, and # *p* < 0.05 compared to LACI.

**Figure 2 ijms-25-09352-f002:**
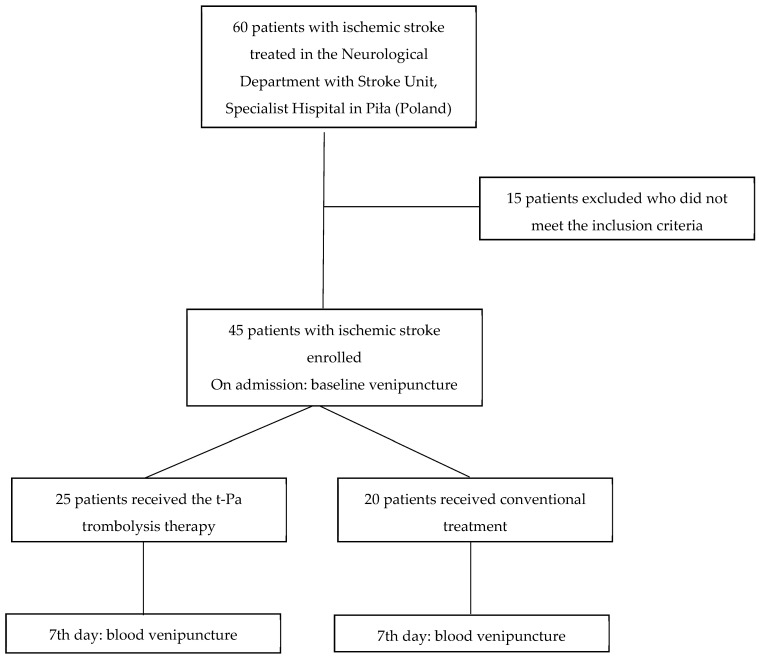
Flow diagram of patient selection and blood venipuncture procedure.

**Table 1 ijms-25-09352-t001:** Pre-treatment demographic and clinical characteristics of the study group.

Variable		Total (n = 45)	Type of Treatment		*p* Value
			Thrombolysis (n = 25)	Conventional (n = 20)	
Age, y		69 (63–75)	71 (62–79)	68 (63–72)	0.37
Male-sex, n (%)		24 (53)	12 (48)	12 (60)	0.42
BMI, kg/m^2^		27 (25–29)	28 (25–30)	27 (24–29)	0.36
WHR		1.09 (0.99–1.19)	1.11 (1.01–1.19)	1.08 (0.99–1.16)	0.24
SBP, mmHg		148 (130–160)	151 (140–160)	143 (130–160)	0.34
DBP, mmHg		80 (79–90)	80 (80–90)	83 (75–90)	0.55
Clinical history n/total (%)					
CAD		8/45 (18)	5/25 (20)	3/20 (15)	0.66
AMI		10/45 (22)	7/25 (28)	3/20 (15)	0.42
AF		12/45 (27)	8/25 (32)	4/20 (20)	0.37
Hypertension		32/45 (71)	19/25 (76)	13/20 (65)	0.30
Diabetes mellitus		12/45 (27)	6/25 (24)	6/20 (30)	0.65
Dyslipidemia		37/45 (82)	22/25 (88)	15/20 (75)	0.26
Smokers		12/45 (27)	6/25 (24)	6/20 (30)	0.65
TOAST	CE	13/45 (29)	9/25 (36)	4/20 (20)	0.20
	LAA	8/45 (18)	3/25 (12)	5/20 (25)	
	SVO	7/45 (15)	2/25 (8)	5/20 (25)	
	SUE	17/45 (38)	11/25 (44)	6/20 (30)	
OCSP	LACI	17/45 (38)	9/25 (36)	8/20 (40)	0.17
	POCI	8/45 (18)	2/25 (8)	6/20 (30)	
	TACI	5/45 (11)	4/25 (16)	1/20 (5)	
	PACI	15/45 (33)	10 (40)	5/20 (25)	
Stroke scales					
NIHSS		6 (4–11)	7 (3–11)	5.5 (4–11)	0.44
mRS		3 (2–4)	3 (2–4)	3 (2–4)	0.50
BI		80 (42.5–90)	80 (40–85)	80 (55–90)	0.22

The data are presented as the median (IQR) for continuous variables and n (%) for categorical variables. Abbreviations: AF, atrial fibrillation; AMI, acute myocardial infarction; BMI, body mass index; CAD, coronary artery disease; CE, cardioembolism; DBP, diastolic blood pressure; LAA, large artery atherosclerosis; LACI, lacunar infarct; PACI, partial anterior circulation infarct; POCI, posterior circulation infarct; SBP, systolic blood pressure; SUE, stroke of undetermined etiology; SVO, small vessel occlusion; TACI, total anterior circulation infarct; and WHR, waist–hip ratio.

**Table 2 ijms-25-09352-t002:** Parameters of iron metabolism in patients with ACS on admission and on the 7th day, depending on the treatment.

		On Admission	7th Day	*p* Value
	Group	Median IQT	Median IQT	
Ferritin, ng/mL	Whole study group	171.40 101.50; 243.60	208.80 151.80; 282.10	**0.03**
	Thrombolysis	173.70 99.16; 228.60	194.10 118,80; 264,10	**0.02**
	Conventional treatment	146.15 106.65; 251.20	215.95 170.00; 285.80	0.47
*p* value *		0.98	0.86	
sTfR, µg/mL	Whole study group	1.31 0.93; 1.75	1.27 1.04; 1.70	0.42
	Thrombolysis	1.44 1.12; 1.88	1.43 1.11; 1.84	0.30
	Conventional treatment	1.21 0.70; 1.60	1.17 0.91; 1.48	0.91
*p* value *		0.14	0.10	
Hepcidin, ng/mL	Whole study group	47.16 29.33; 84.81	37.88 19.96; 72.09	**0.04**
	Thrombolysis	58.43 29.66; 118.40	48.82 27.03; 66.12	0.08
	Conventional treatment	44.12 22.96; 64.88	30.76 8.33; 76.00	0.24
*p* value *		0.24	0.32	
sHJV, pg/mL	Whole study group	1862.00 1088.00; 2539.00	1609.00 1140.00; 2136.00	0.21
	Thrombolysis	2468.00 1962.00; 2972.00	2067.00 1609.00; 2342.00	0.10
	Conventional treatment	1115.50 916.45; 1463.50	1175.50 794.30; 1383.50	0.97
*p* value *		**<0.0001**	**0.0002**	

* thrombolysis vs. conventional treatment.

**Table 3 ijms-25-09352-t003:** Comparison of iron metabolism parameters depending on stroke severity.

	NIHSS < 8 n = 27	NIHSS 8–15 n = 13	NIHSS ≥ 16 n = 5	*p* Value
	Median IQT	Median IQT	Median IQT	
Ferritin, ng/mL on admission	195.40 110.60; 310.50	141.80 95.76; 191.70	204.10 120.00; 211.30	0.32
Ferritin, ng/mL 7th day	234.30 158.70; 345.30	169.60 132.00; 193.20	250.00 170.30; 414.90	0.07
sTfR, µg/mL on admission	1.24 0.82; 1.72	1.31 0.84; 1.92	1.50 1.44; 1.51	0.34
sTfR, µg/mL 7th day	1.27 1.11; 1.69	1.04 0.76; 1.61	1.84 1.58; 1.88	0.19
Hepcidin, ng/mL on admission	50.57 27.75; 79.25	42,86 29.66; 84.81	82.72 47.13; 141.30	0.75
Hepcidin, ng/mL 7th day	38.77 16.05; 89.13	30.63 25.51; 51.34	51.34 33.81; 67.45	0.68
sHJV, pg/mL on admission	1682.00 970.30; 2539.00	1861.00 1455.00; 2468.00	2238.00 1287.00; 2968.00	0.64
sHJV, pg/mL 7th day	1426.00 1059.00; 2067.00	1900.00 1140.00; 2327.00	1759.00 1314.00; 2122.00	0.86

**Table 4 ijms-25-09352-t004:** Comparison of iron metabolism parameters depending on the etiology of stroke.

	Stroke Subtype				
	Undetermined Etiology n = 17	Cardioembolism n = 13	Small-Vessel Occlusion n = 7	Large-Vessel Occlusion n = 8	*p* Value
	Median IQT	Median IQT	Median IQT	Median IQT	
Ferritin, ng/mL on admission	171.40 101.50; 204.10	141.80 95.69; 228.60	211.20 95.76; 310.50	205.20 133.25; 258.95	0.82
Ferritin, ng/mL 7th day	208.80 169.60; 234.30	151.80 104.10; 345.30	192.30 132.00; 282.10	299.10 179.60; 382.50	0.54
sTfR, µg/ml on admission	1.51 1.15; 1.82	1.38 1.22; 2.02	0.93 0.72; 1.70	0.86 0.66; 1.38	**0.02 ***
sTfR, µg/mL 7th day	1.32 0.98; 1.70	1.25 1.11; 2.13	1.27 0.76; 1.52	1.28 0.79; 1.63	0.69
Hepcidin, ng/mL on admission	50.57 29.37; 96.37	34.36 10.33; 67.35	47.16 32.12; 79.25	65.99 30.58; 129.85	0.54
Hepcidin, ng/mL 7th day	38.77 27.03; 67,45	30.63 19.96; 66.12	31.85 2.98; 50.74	70.33 29.44; 107.70	0.40
sHJV, pg/mL on admission	1962.00 1424.0; 2468.0	2449.00 1523.0; 2972.0	1398.00 883.1; 1682.1	1115.50 973.8; 2423.3	0.11
sHJV, pg/mL 7th day	1609.00 1258.0; 2277.0	1900.00 1303.0; 2375.0	1059.00 745.4; 1303.0	1532.92 1257.0; 1891.0	0.13

* significant difference in sTfR concentration between undetermined etiology and large-vessel occlusion.

**Table 5 ijms-25-09352-t005:** Comparison of iron metabolism parameters depending on the stroke location.

	LACI n = 17	POCI n = 8	TACI n = 5	PACI n = 15	*p* Value
	Median IQT	Median IQT	Median IQT	Median IQT	
Ferritin, ng/mL on admission	147.20 110.60; 282.60	192.45 153.00; 284.65	141.80 92.19; 206,30	171.40 69.43; 208.10	0.33
Ferritin, ng/mL 7th day	194.10 170.30; 368.70	259.95 198,60; 327,00	169.60 151.80; 350.10	193.20 107.00; 234.30	0.24
sTfR, µg/mL on admission	1.37 0.82; 1.70	0.98 0.68; 1.34	1.88 1.26; 3.03	1.40 1.15; 1.92	0.12
sTfR, µg/mL 7th day	1.26 0.89; 1.69	1.17 0.86; 1.32	1.71 1.62; 3.22	1.32 0.98; 1.74	0.09
Hepcidin, ng/mL on admission	45.25 27.75; 111.00	58.80 45.08; 66.00	13.32 10.33; 118.40	50.57 29.66; 84.81	0.91
Hepcidin, ng/mL 7th day	33.81 16.05; 61.54	63.04 22.49; 81.33	25.51 23.59; 48.82	37.88 27.03; 67.45	0.74

## Data Availability

The data presented in this study are available from the corresponding author upon reasonable request.

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
