# Peer review of "Insights into Iron Metabolism Parameters in Ischemic Stroke: A Single-Center Prospective Cohort Study"

_ijms, 2024, doi:10.3390/ijms25179352_

Round 1

Reviewer 1 Report

Comments and Suggestions for Authors

This paper demonstrates a short time course follow up of certain ischemic stroke patients considering a few of their iron related blood parameters. The 45 patients are grouped according to the location, etiology and treatment of their disease. Hepcidin, sTR, ferritin, and sHJV levels were determined, and relationships were sought between iron metabolism and the features of stroke. The authors found some promising connections in their research.

The paper is clearly written, the methods used are correct. Background information, interpretation of the data, statistical methods and references are acceptable.

My questions: What is the conventional treatment of the patients? Heparin is mentioned but no other drugs or methods. What is the reason that these four parameters were checked? Did the authors measure other iron related parameters, but do not show them in the results? In the introduction macrophages should be mentioned also, they play relevant role in iron metabolism. In line 270 there is a repeated sentence. Do the authors think that their results can be used in therapy or diagnosis of stroke?

Reviewer 2 Report

Comments and Suggestions for Authors

The authors measured iron metabolism biomarkers in Ischemic Stroke (SI) patients and evaluated the effects of treatment with thrombolysis.

Major concerns:

The statement that: Our findings confirm that changes in the hepcidin-sHJV axis are involved in the pathophysiology of IS is unjustified in light of the presented results. The mere observation of increased sHJV levels in patients with PACI stroke and a decrease in Herpcidin does not prove a causal relationship. Too few patients and an attempt to translate the observed phenomenon in peripheral blood with CNS circulation and correlation with stroke is an overinterpretation.

Did the decrease in peripheral blood hepcidin entail changes in serum iron levels?

The main donor of sHJV in serum is muscle, so how can we explain the situation after stroke?

If the authors postulate interaction/regulation of the sHJV-Hepc axis, why do they write in the discussion: Our research observed that hepcidin concentration significantly decreases on the seventh day of hospitalization, regardless of the treatment administered during the acute  phase of stroke. This decrease is likely due to the subsiding inflammatory process and the reduction of inflammatory mediators, which are primary stimulators of hepcidin synthesis. It seems that it’s not a case.

Table 2: On admission, the results of sHJV in the Thrombolysis and Conventional treatment groups show significant differences. Other parameters, such as hepcidin and ferritin, also show differences in values, although these are not significant (likely due to large data errors). This indicates that baseline of the groups was uneven before the thrombolysis and conventional treatments.

Table 3, 4, and 5: The authors only compared the differences in the "Whole study group," but ignored the comparisons of "on admission" and "7th day," as well as the comparison between "Thrombolysis" and "Conventional treatment."

Minor concerns:

Table 2: Missing sTfR data in the conventional treatment group on the 7th day.

Table 2, 3, and 5: Ferritin fonts are different.

Table 2: Hepcidin table formatting is incorrect.

Is there a numbering error in "2.4, 2.5, 2.6"? It should be "2.3.1, 2.3.2, 2.3.3"’.

Figure 1: Why the sHJV indicator is a separate figure instead of being included in Table 5?.

Line 146, 171-181: The abbreviation "HJV" for hemojuvelin is not used here. Once the abbreviation is defined in the main text, it should be used consistently thereafter.

Line 152-169: In the second to fourth paragraphs, the authors did not expand on the discussion, but only re-described the experimental results, which is unnecessary.

Line 160: "The results were significant...", but in fact, only sHJV was found with a difference, while other indicators did not change.

Line 185-187: "Elevated levels of sHJV reduce the expression of hepcidin", but hepcidin did not change in this experiment. The author should focus on discussing the data presented in this paper.

Lines 199-226: The discussion here is excessive and lacks sufficient references. "This decrease is likely due to the subsiding inflammatory process...", however the authors did not detect any inflammation-related indicators.

Line 229: "These findings are in line with the studies conducted by other authors" lacks references.

Line 234: "The research ... a cognitive nature." This sentence is confusing.

The abbreviation for “coefficient of variation (CV)" appears twice.

Comments on the Quality of English Language

no

Round 2

Reviewer 2 Report

Comments and Suggestions for Authors

Dear authors, thank you for taking the time to respond to my concerns and comments. I believe that in this form and with these final conclusions the article is acceptable for publication.

Best regards.